# Hepatic Histopathological Benefit, Microbial Cost: Oral Vancomycin Mitigates Non-Alcoholic Fatty Liver Disease While Disrupting the Cecal Microbiota

**DOI:** 10.3390/ijms26178616

**Published:** 2025-09-04

**Authors:** Gül Çirkin, Selma Aydemir, Burcu Açıkgöz, Aslı Çelik, Yunus Güler, Müge Kiray, Başak Baykara, Ener Çağrı Dinleyici, Yeşim Öztürk

**Affiliations:** 1Department of Pediatric Gastroenterology Hepatology and Nutrition, Faculty of Medicine, Dokuz Eylul University, Izmir 35340, Türkiye; dryunusguler@gmail.com (Y.G.); y.zaferozturk@gmail.com (Y.Ö.); 2Department of Histology and Embryology, Faculty of Medicine, Dokuz Eylul University, Izmir 35340, Türkiye; selma.aydemiir@gmail.com (S.A.); basakguzelbaykara@gmail.com (B.B.); 3Department of Physiology, Faculty of Medicine, Dokuz Eylul University, Izmir 35340, Türkiye; buracikgoz@gmail.com (B.A.); muge.kiray@deu.edu.tr (M.K.); 4Experimental Animals Laboratory, Faculty of Medicine, Dokuz Eylul University, Izmir 35340, Türkiye; vet.asli.celik@gmail.com; 5Department of Pediatrics, Faculty of Medicine, Eskisehir Osmangazi University, Eskisehir 26040, Türkiye; enercagri@gmail.com

**Keywords:** oral vancomycin, non-alcoholic fatty liver disease, liver inflammation, mice ubiquitin, cytokeratin 18, microbiota

## Abstract

Non-alcoholic fatty liver disease (NAFLD) and non-alcoholic steatohepatitis (NASH) encompasses a spectrum of liver conditions and involves gut–liver axis crosstalk. We aimed to evaluate whether oral vancomycin modifies liver injury and the cecal microbiota in a methionine–choline-deficient (MCD) diet model of NASH. Male C57BL/6J mice (n = 28) were block-randomized to four groups (n = 7 each) for 10 weeks: standard diet (STD); MCD diet; STD + vancomycin (VANC); and MCD + VANC (2 mg/mouse ≈ 50 mg/kg, every 72 h). After 10 weeks, liver tissues were analyzed for histological changes, cytokine levels [interleukin-6 (IL-6), interleukin-8 (IL-8), transforming growth factor beta 1 (TGF-β1)], and immunohistochemical markers [ubiquitin and cytokeratin 18 (CK18)]. Cecal microbiota composition was evaluated with 16S ribosomal RNA (rRNA) sequencing. The MCD reproduced key NASH features (macrovesicular steatosis, lobular inflammation). Vancomycin shifted steatosis toward a microvesicular pattern and reduced hepatocyte injury: CK18 and ubiquitin immunoreactivity were decreased in MCD + VANC vs. MCD, and hepatic IL-8 and TGF-β1 levels were lower in MCD + VANC vs. STD. Taxonomically, STD mice had *Lactobacillus*-rich microbiota. The MCD diet alone reduced alpha diversity (α-diversity), modestly lowered Firmicutes and increased *Desulfobacterota*/*Fusobacteriota*. Vancomycin alone caused a much larger collapse in richness, depleting Gram-positive commensals and promoting blooms of *Escherichia–Shigella*, *Klebsiella*, *Parabacteroides*, and *Akkermansia*. In the MCD + VANC group, vancomycin profoundly remodeled the microbiota, eliminating key commensals (e.g., *Lactobacillus*) and enriching *Desulfobacterota*, *Fusobacteriota*, and *Campylobacterota*. Oral vancomycin in the MCD model of NASH improved liver injury markers and altered steatosis morphology, but concurrently reprogrammed the gut into a low-diversity, pathobiont-enriched ecosystem with near-loss of *Lactobacillus*. These findings highlight a therapeutic trade-off—hepatic benefit accompanied by microbiome cost—that should guide microbiota-targeted strategies for NAFLD/NASH.

## 1. Introduction

Non-alcoholic fatty liver disease (NAFLD) encompasses a wide spectrum of liver conditions, from simple steatosis (fat accumulation in hepatocytes) to non-alcoholic steatohepatitis (NASH), fibrosis, cirrhosis, and hepatocellular carcinoma. It is the most common chronic liver disease, affecting ~25% of the global population [1,2,3]. The proposed term metabolic dysfunction-associated fatty liver disease (MAFLD) emphasizes the close link between metabolic derangements and liver injury [2]. Although obesity and hyperlipidemia are primary contributors to NAFLD, several systemic and inflammatory conditions and hepatotoxic drugs or nutrient deficiencies can precipitate hepatic steatosis [4]. Dysbiosis in individuals with NAFLD is linked to abnormal regulation of the liver–gut axis, gut microbial constituents, and microbial metabolites [5]. Modulating the gut microbiota has been suggested as a potential therapeutic strategy for the treatment of NAFLD [5,6]. Microbial compounds (e.g., lipopolysaccharide), bile acid remodeling, and short-chain fatty acids affect hepatic inflammation and lipid metabolism [7].

NAFLD is increasingly recognized in children and adolescents, corresponding to a rise in obesity and metabolic syndrome, highlighting the urgent necessity for appropriate treatment strategies [8]. Despite progress in comprehending the fundamental causes of NAFLD, treatment options remain limited. This therapeutic gap has driven researchers to explore novel approaches targeting the gut–liver axis [9]. Oral vancomycin has demonstrated efficacy in immune-mediated hepatobiliary disorders (e.g., primary sclerosing cholangitis), autoimmune liver disease, and ulcerative colitis, possibly by altering intestinal microbiota and bile acid composition [10,11,12]. In humans, vancomycin treatment diminished fecal microbial diversity, lowered secondary bile acids, and reduced peripheral insulin sensitivity in individuals with metabolic syndrome [11]. However, antibiotic effects in metabolic liver disease are context-dependent; depleting Gram-positive commensals may also permit expansion of other pathobionts that may worsen hepatic injury.

To interrogate this, we used the methionine–choline-deficient (MCD) diet, a well-established murine model that rapidly induces the histologic hallmarks of NASH (steatosis, lobular inflammation, ballooning, and early fibrosis), albeit without obesity or insulin resistance [9,13]. We superimposed oral vancomycin to test whether antibiotic-driven microbiota modulation alters MCD-induced liver injury. Therefore, the objective of this study was to investigate the effects of superimposed oral vancomycin on histopathological changes, including liver fibrosis, inflammatory markers, and gut microbiota composition in an experimental NAFLD model.

## 2. Results

### 2.1. Body Weight

Across the 10-week study, no difference in body weight was observed among groups (*p* > 0.05) (Appendix A).

### 2.2. Liver Histological Findings

STD Group: Liver tissue exhibited normal hepatocyte morphology, well-defined hepatic cords, intact sinusoids, and central veins. PAS and Masson’s trichrome were unremarkable for glycogen accumulation and collagen distribution (Figure 1 and Figure 2).

MCD Group: We observed significant pathological changes, including vacuolar degeneration of hepatocytes, irregular hepatic cords, and dilated sinusoids. Notable lobular inflammation and macrovesicular steatosis were observed, together with an increase in eosinophilic cells. Glycogen staining decreased in comparison to STD; collagen distribution was largely preserved (Figure 1 and Figure 2).

VANC Group: We observed pronounced lobular infiltration, dilated vena centralis structures, increased eosinophilic cells, irregular hepatic cords, and dilated sinusoids. Glycogen and collagen patterns resembled those in MCD (Figure 1 and Figure 2).

MCD + VANC Group: Livers displayed a shift toward microvesicular steatosis compared with MCD alone, reduced vacuolar degeneration and sinusoidal dilatation, and improved cellular morphology relative to both the MCD and VANC groups (Figure 1 and Figure 2). Collagen staining was similar across groups; glycogen was higher than in MCD and VANC.

Semi-quantitative scoring. NAS components (steatosis, lobular inflammation, and ballooning) increased with MCD; the combined group showed reduced steatosis grade relative to MCD but persistent inflammation (Figure 2).

### 2.3. Immunohistochemistry

Cytokeratin-18 (CK18)-positive hepatocytes were concentrated around central veins in the MCD and MCD + VANC groups. In both groups, reactivity in the portal area was low. CK18 immunoreactivity severity was lower in MCD + VANC than MCD. In the STD and VANC groups, reduced immunoreactivity was observed in the central vein and portal areas, with no significant difference between these two groups (Figure 2 and Figure 3).

Ubiquitin immunoreactivity was strongest in the MCD and VANC groups (hepatocytes and inflammatory cells). The MCD group showed more severe ubiquitin immunoreactivity than the STD, VANC, and MCD + VANC groups. No significant difference in ubiquitin immunoreactivity severity was noted between the MCD + VANC and STD groups (Figure 2 and Figure 3).

### 2.4. Hepatic Cytokine Findings

Mean values of hepatic cytokines are shown in Table 1. IL-6 did not differ among groups. IL-8 varied overall (*p* = 0.023) and was lower in MCD + VANC vs. STD (*p* = 0.003); there was no significant difference between the MCD and STD. IL-14 differed across groups (*p* = 0.005) and was lower in MCD + VANC vs. STD (*p* = 0.009). TGF-β1 also differed between groups (*p* = 0.045) and was lower in MCD + VANC vs. STD (*p* = 0.006).

### 2.5. Microbiota Analysis

#### 2.5.1. Alpha-Diversity

Among the study groups, the lowest Shannon index was observed in the MCD + VANC group. Both MCD and vancomycin reduced Shannon diversity relative to STD, with the lowest diversity in the combined MCD + VANC group, which did not differ significantly from vancomycin alone. The pairwise comparisons were as follows: MCD vs. STD, *p* < 0.05; VANC vs. STD, *p* < 0.001; MCD + VANC vs. STD, *p* < 0.001; MCD + VANC vs. MCD, *p* = 0.003; MCD + VANC vs. VANC, *p* = 0.176. Vancomycin, either alone or combined with the MCD diet, reduced the Chao1 index dramatically (VANC = 93 ± 12 vs. STD = 546 ± 38, *p* < 0.001). MCD diet alone produced a moderate reduction (315 ± 45 vs. STD = 546 ± 38; *p* < 0.001). Both MCD and vancomycin markedly reduced Chao1 richness relative to STD, with the lowest richness in the combined MCD + VANC group, which did not differ significantly from vancomycin alone (MCD + VANC = 81 ± 9 vs. VANC = 93 ± 12, ns). Simpson evenness was unchanged by either the MCD diet or vancomycin alone but fell significantly when the two were combined (*p* < 0.01) (Figure 4a,b). 

#### 2.5.2. Beta-Diversity

Bray–Curtis PCoA showed clear segregation of all four groups along two orthogonal axes: PC1 (30%) reflected the MCD diet, whereas PC2 (28%) captured vancomycin exposure (Figure 5a). Two-way PERMANOVA confirmed significant main effects of diet (R^2^ = 0.31, q < 0.01) and antibiotics (R^2^ = 0.28, q < 0.01) plus a smaller but significant MCD × vancomycin interaction (R^2^ = 0.07, q = 0.02), indicating that vancomycin modifies the dietary impact on gut community structure.

Presence/absence-based Jaccard PCoA (Figure 5b) explained 41% of variation in the first two axes. PC1 (28%) cleanly separated antibiotic-treated from untreated mice, while PC2 (13%) distinguished the combined MCD + VANC group from all others. Two-way PERMANOVA confirmed significant main effects of vancomycin and MCD as well as a diet × antibiotic interaction.

#### 2.5.3. Cecal Microbiota Composition

Venn analysis of amplicon sequence variants (ASVs) underscored the stepwise loss and replacement of taxa across interventions. The standard diet supported 1315 detectable ASVs; the MCD diet removed roughly half of these taxa (total 565 ASVs), whereas vancomycin culled > 85% of baseline diversity (188 ASVs). Intriguingly, combining the MCD diet with vancomycin yielded 347 ASVs, including 264 not found in any other group, indicating that the antibiotic-cleared niche was re-colonized by diet-adapted, vancomycin-resistant organisms.

At the phylum level, the STD group displayed a typical Firmicutes/Bacteroidota-dominated community (≈50%/45%). The MCD diet modestly lowers Firmicutes and increases Desulfobacterota/Fusobacteriota; vancomycin produces a marked expansion of Proteobacteria with reductions in Firmicutes and Bacteroidota and a small Verrucomicrobiota band. Vancomycin treatment in the MCD group reshapes the community with prominent Desulfobacterota, Fusobacteriota, and Campylobacterota and persistently low Firmicutes and lower Proteobacteria (comparing vancomycin-alone group, Figure 6).

At the genus level, in the STD Group, *Lactobacillus* (~25%) was most abundant, followed by *Lachnospiraceae_NK4A136_group* (3.1%), *Odoribacter* (2.9%), *Alistipes* (2.3%), *Helicobacter* (2.0%), *Bacteroides* (1.4%), *Brachyspira* (1.1%), and *Rikenella* (1.6%). Conversely, *Lactobacillus* declined in all other groups. In the MCD group, loss of *Lactobacillus* (0.24%) and emergence of diet-adapted taxa were observed: *Colidextribacter* 10.4%, *Helicobacter* 9.3%, *Brachyspira* 3.3%, *Bacteroides* 2.4%, and Blautia 3.0%. Genus-level profiling exposed marked blooms of putative pathobionts following vancomycin treatment. In the VANC group, Proteobacteria bloom with *Escherichia*–*Shigella* at 20.1%, *Parabacteroides* at 13.4%, *Akkermansia* at 11.4%, *Fusobacterium* at 7.5%, *Klebsiella* at 4.0% and *Lactobacillus* falls to 3.2%. In the MCD + VANC group, a distinct pathobiont-dominated consortium included *Fusobacterium* (25.9%), *Helicobacter* (10.4%), *Escherichia–Shigella* (6.5%). *Ligilactobacillus* (4.2%), *Veillonella* (1.1%), *Lachnoclostridium* (1.2%), *Desulfovibrio* (1.3%), *Lactococcus* (0.45%) were also present. *Lactobacillus* was nearly absent (0.005%) (Figure 7). Genus-level heatmaps of the cecal microbiota across diet/antibiotic groups are shown in Figure 8.

#### 2.5.4. Differential Abundance Snapshot

STD vs. MCD diet

Compared with standard chow, the MCD diet had lower *Lactobacillus* (*p* < 0.01), while enriched in *Helicobacter*, *Coliddextribacter*, *Oscillibacter*, and *Paludicola* (Appendix A).

STD vs. VANC

Vancomycin depleted key Gram-positive commensals, with significant decline in *Lactobacillus* and depletion of *Bacteroides* and *Alistipes*. Vancomycin promotes blooms of *Escherichia-Shigella* and *Parabacteroides* and *Akkermansia*, *Parasutterella*, *Erysipelatoclostridium*, and *Paludicola*. These shifts align with the Proteobacteria-dominated phylum profile and the marked loss of taxonomic richness observed after vancomycin treatment (Appendix A).

STD vs. MCD + VANC

Superimposing vancomycin on the MCD diet eliminated key commensals (e.g., *Lactobacillus*) while promoting blooms of *Fusobacterium*, *Escherichia-Shigella*, *Parasutterella*, and the sulfate-reducer *Desulfovibrio*, defining a dysbiotic community uniquely associated with the combined treatment (Appendix A).

MCD vs. VANC

Comparing the MCD diet group to vancomycin treatment revealed distinct, non-overlapping ecological signatures. MCD selectively enriched *Helicobacter*, *Bacteroides*, and *Rikenellaceae RC9*, whereas vancomycin favored Gram-negative taxa, notably *Escherichia-Shigella*, *Parabacteroides*, and *Akkermansia* (Appendix A).

MCD vs. MCD + VANC

Superimposing vancomycin on the MCD diet profoundly remodeled the residual microbiota. A marked bloom of *Fusobacterium*, *Desulfovibrio*, and *Escherichia-Shigella* was detected, concurrent with the loss of *Brachyspira* that was enriched by MCD alone (Figure 9a,b).

VANC vs. MCD + VANC

In vancomycin-treated mice, *Escherichia-Shigella*, *Akkermansia*, and *Parabacteroides* were dominant, while the pathobiont consortium was dominated by *Fusobacterium* and sulfate-reducing Desulfovibrio in the MCD-VAC group (Figure 10a,b).

#### 2.5.5. LEfSe Multiclass Analysis

LEfSe multiclass analysis (Figure 11) identified distinct biomarker consortia for each intervention. The STD group were characterized by SCFA-producing *Lactobacillus* (-aceae and specific ASVs) and *Bifidobacteriaceae* (LDA > 4). *Odoribacter*, *Lachnospiraceae* NK4A136, and *Clostridium UCG-014* were also abundant. In the MCD group, top discriminant taxa were *Lachnospiraceae bacterium 28-4*, *Oscillospiraceae*, *Ruminococcus*, *Clostridiales vadinBB60*, and *Muribaculaceae*. The MCD diet enriched mucus-associated *Oscillospiraceae*, whereas combining the MCD diet with vancomycin produced a unique pathobiont triad: *Fusobacterium, Helicobacter ganmani*, and *Desulfovibrio* (all LDA > 4). Vancomycin treatment caused a Proteobacteria-heavy signature, led by *Gammaproteobacteria*, *Enterobacteriaceae*, *Escherichia-Shigella*, and *Klebsiella* (LDA > 4.5). Tenericutes/Verrucomicrobia phyla (incl. Akkermansia), *Parabacteroides*, and *Barnesiellaceae* were also abundant. In the MCD + VANC group, *Fusobacterium*, *Helicobacter ganmani*, *Desulfovibrionaceae*/*Desulfovibrio*, *Campylobacteraceae*, and *Fusobacteriaceae* were abundant.

## 3. Discussion

In this experimental study, the MCD diet reproduced core histologic features of NASH (macrovesicular steatosis and lobular inflammation), whereas superimposed oral vancomycin shifted steatosis toward a microvesicular pattern and reduced hepatocyte injury readouts. Specifically, CK18 and ubiquitin immunoreactivity were lower in MCD + VANC compared with MCD, and hepatic IL-8 and TGF-β1 were reduced versus the standard diet, while body weight was unchanged. These findings indicate that vancomycin modifies liver injury signals in the MCD model.

We chose to use the MCD model rather than conduct a high-fat-diet study of gut–liver interactions independent of obesity. Unlike a high-fat diet, which induces systemic inflammation, increases gut permeability, and causes proinflammatory macrophage activation in white adipose tissue, the MCD diet elicits liver steatosis and inflammation without the confounding effect of obesity [14,15]. The macrovesicular steatosis detected in the MCD group indicates advanced hepatocellular fat deposition, typically associated with progressive stages of NAFLD. In contrast, a predominance of microvesicular steatosis was noted in the MCD + VANC group, suggesting a decrease in steatosis severity. This histological shift from macro- to microvesicular fat deposition implies a possible protective effect attributable to oral vancomycin.

The results of this study showed that CK18 and ubiquitin immunoreactivity were significantly elevated in the MCD group, consistent with their roles in inflammation, apoptosis, and proteasomal stress as biomarkers of hepatocyte injury [16,17,18,19], and reduced in MCD + VANC, compatible with a hepatoprotective signal. The decrease in CK18 levels in MCD + VANC group indicates reduced apoptosis and inflammation, while the reduction in ubiquitin levels suggests partial restoration of proteasomal function and decreased cellular stress. Together with significantly lower levels of proinflammatory cytokines IL-8 and TGF-β in the MCD + VANC group compared to other groups, these changes indicate an anti-inflammatory/anti-fibrogenic profile in vancomycin-treated MCD livers. IL-8 was correlated with inflammatory and fibrotic changes, which are associated with NAFLD progression [20,21]. Similarly, TGF-β is recognized as a pivotal cytokine in fibrosis development, and the significantly lower TGF-β1 levels in the MCD + VANC group indicate a possible role for vancomycin in attenuating fibrotic responses together with its anti-inflammatory effects [21].

There are limited studies about the relationship between gut microbiota, vancomycin treatment, and liver diseases, particularly NAFLD and NASH [22,23]. Vancomycin treatment has been shown to exacerbate liver damage, steatohepatitis, and fibrosis in mice fed an iHFC diet (high-fat/cholesterol/cholate-based diet) [22]. Kasai et al. [22] showed that vancomycin worsened the progression of liver damage, steatohepatitis, and fibrosis, and they concluded that vancomycin aggravates iHFC-NASH through microbiota-dependent bile acid shifts and macrophage-driven inflammation, consistent with loss of Gram-positive bile acid-modifying taxa [22]. Takano et al. [23] highlighted the impact of gut microbiota on the dynamics of liver macrophages and bile acid metabolism during NASH onset and progression. They evaluated the link between iHFC diet-induced NASH progression and gut microbiota in C57BL/6 mice using antibiotic treatments and showed vancomycin exacerbated the progression of liver damage, steatosis, and fibrosis. The expression levels of inflammation- and fibrosis-related genes in the liver significantly increased after vancomycin treatment for 8 weeks. They showed a decrease in α-diversity in the vancomycin-treated group, with *Bacteroidetes* and *Firmicutes* significantly decreased, while *Proteobacteria* and *Verrucomicrobia* increased markedly. While we observed some potential beneficial effects of vancomycin on histopathological findings related to NASH, microbiota analyses show opposite findings. The MCD diet alone reduced α-diversity (Shannon ↓ ~25%; Chao1 ↓ ~40%) but preserved evenness, modestly lowered *Firmicutes,* and increased *Desulfobacterota/Fusobacteriota*. Vancomycin alone caused a much larger drop in richness (Chao1 ↓ >80%) than the MCD diet alone with a classic antibiotic signature: loss of Gram-positive commensals and blooms of *Escherichia–Shigella*, *Klebsiella*, *Parabacteroides*, and *Akkermansia*. In the MCD group, α-diversity remained low during vancomycin treatment. Superimposing vancomycin on the MCD diet profoundly remodeled the residual microbiota, eliminating key commensals (e.g., *Lactobacillus*) while promoting *Desulfobacterota*, *Fusobacteriota*, and *Campylobacterota*. Increased abundance of *Fusobacterium*, *Escherichia-Shigella*, *Parasutterella*, and the sulfate-reducer *Desulfovibrio* was observed, indicating a dysbiotic community uniquely associated with the combined treatment. We hypothesize that “vancomycin rescues NASH by reshaping the microbiome,” and the sharp fall in the Shannon score indicates that the intervention was effective. Whether the direction of that shift is hepatoprotective is another matter; extreme richness loss can permit a bloom of pathobionts that translocate lipopolysaccharide or ethanol, potentially aggravating steatohepatitis. The MCD milieu determines which survivors expand, and after vancomycin treatment, organisms mechanistically linked to hepatotoxic pathways—enteric LPS, hydrogen sulfide production by *Desulfovibrio*, and mucus erosion/epithelial perturbation by *Helicobacter/Fusobacterium*—become dominant. Thus, the hepatic signal improvement (↓ CK18/ubiquitin; ↓ IL-8/TGF-β1) likely reflects specific taxonomic/metabolic shifts or bile acid modulation, rather than any increase in global diversity. Overall, our data highlight a potential therapeutic trade-off of liver benefit accompanied by microbiome cost, leading to a low-richness, low-evenness community enriched in pathobionts. Kasai et al.’s study [22], Takano et al.’s study [23], and our study show vancomycin profoundly reduces α-diversity/richness, depletes Gram-positive commensals (e.g., *Lactobacillus*), and blooms *Proteobacteria*. These findings identify diet × antibiotic interactions that create a distinct community rather than a simple additive effect. Regarding the diet context, iHFC contains cholate and strongly perturbs bile acid pools and the gut barrier; MCD drives lipotoxicity without intestinal barrier integrity or intestinal inflammation [24]. Vancomycin on iHFC removes bile acid-modifying Gram-positives, resulting in macrophage-driven fibrosis [22,23]. In MCD, vancomycin still reduces richness but—without the cholate context—can lower hepatic injury markers despite dysbiosis. Unlike iHFC–TSNO studies where vancomycin aggravates cholate-driven fibrosis via bile acid disruption, our MCD–C57BL/6J experiment shows reduced CK18/ubiquitin and pro-fibrotic cytokines despite profound dysbiosis, underscoring that the hepatic consequences of antibiotic-driven microbiome shifts are diet-context-dependent.

The reduction in IL-6 and IL-8 in the MCD-VANC groups suggests a dampening of hepatic inflammation, while decreased TGF-β1 levels are compatible with reduced fibrogenesis. This cytokine profile parallels the histological improvements observed. However, vancomycin induces profound dysbiosis, including the enrichment of pathobionts, which means that the protective cytokine changes may coexist with an increased risk of microbial translocation and endotoxin-driven injury. This duality underscores that while vancomycin can reduce inflammatory mediators, its long-term benefit for liver health is uncertain unless paired with microbiota-supportive strategies. The apparent paradox of lower IL-6/IL-8/TGF-β1 levels alongside the highest inflammatory profile (lobular inflammation) in the VANC alone group underscores the complex interplay between the microbiota and host responses. Vancomycin depleted key Gram-positive commensals, leading to a significant decrease in Lactobacillus and a reduction in Bacteroides. Vancomycin promotes blooms of Escherichia-Shigella and Parabacteroides. These shifts align with the Proteobacteria-dominated phylum profile and the marked loss of taxonomic richness observed after vancomycin treatment. One possible explanation is that while vancomycin reduces direct hepatic inflammatory signaling, the resulting dysbiosis promotes leaky gut and endotoxin exposure, sustaining systemic and mucosal inflammation. Thus, vancomycin’s net effect reflects both transient liver improvement and heightened inflammatory pressure from the gut environment.

In keeping with a dysbiotic shift, *Lactobacillus*—dominant under a standard diet (~24%)—was markedly depleted by both interventions and nearly ablated when vancomycin was combined with the MCD diet. Given Lactobacillus’ roles in bile-salt hydrolysis, SCFA production, and epithelial barrier support [25,26], its depletion plausibly lowers colonization resistance and favors *Proteobacteria/Fusobacteria* expansion. These patterns suggest that vancomycin’s broad suppression of Gram-positive commensals—as amplified by the MCD nutrient milieu—removes a protective guild that normally buffers LPS/H_2_S exposure to the liver. Follow-up studies with a probiotic/consortium rescue targeting Lactobacillus with vancomycin would test whether restoring *Lactobacillus* mitigates the microbiota cost while preserving hepatic benefit. Because vancomycin drives richness loss while MCD dictates survivor selection, timing/dose optimization and mitigation strategies (e.g., narrow-spectrum agents or post-antibiotic probiotics) may preserve hepatic benefit while limiting dysbiosis.

The strengths of our study include the factorial design, blinded histology and microbiota analysis, and multi-layer microbiota readouts. Limitations are inherent to the MCD model (lack of obesity/insulin resistance), the use of only male C57BL/6J mice, and a modest sample size. Another limitation of the study is the lack of bile acid, LPS, and permeability marker assessments. Future studies would benefit from including bile acid profiling, measurement of circulating LPS, and functional assays of intestinal permeability, which would provide a more comprehensive understanding of host–microbiota interactions and barrier integrity.

## 4. Materials and Methods

This study is a prospective, parallel-group, interventional study designed to evaluate the effect of adding oral vancomycin on histological findings and microbiota composition in an experimental NASH model. This study was approved by the Animal Experiments Ethics Committee of Dokuz Eylul University (Protocol Number: 48/2021) and supported by the Scientific Research Projects Coordination Unit of Dokuz Eylul University (Project Number: TKB-2022-2664). Experiments followed national regulations and the NIH Guide for the Care and Use of Laboratory Animals (8th ed.) [27]. Mice were block-randomized by body weight into the four groups (n = 7/group). All samples and slides were labeled with anonymized codes; investigators performing histological scoring, immunohistochemistry, cytokine assays, and microbiota preprocessing/analyses were blinded to group allocation until after the primary analyses were completed.

The primary aim of this study was to determine whether oral vancomycin modifies the severity of MCD-induced NASH in C57BL/6J mice (histology and liver injury markers), characterize the antibiotic- and diet-dependent remodeling of the cecal microbiota (α/β-diversity and differential taxa), and relate these patterns to liver pathology.

### 4.1. Animals

Male C57BL/6J mice of 8 to 10 weeks of age were housed under SPF conditions (12 h light/dark, 20–22 °C, 40–60% humidity) with ad libitum access to standard chow and water. Bedding was changed regularly to maintain hygienic conditions, and animals were monitored daily; no unexpected deaths occurred during the study.

### 4.2. Experimental Groups and Numbers

Mice (n = 28) were randomly assigned to groups, with 7 mice per group, and followed up for 10 weeks:STD group (n = 7): Mice were fed a standard diet (STD) and water and administered 1 mL of serum physiological (SP) via oral gavage.MCD group (n = 7): Mice were fed a methionine–choline-deficient (MCD) diet and water and administered 1 mL of SP via oral gavage.VANC group (n = 7): Mice were fed a standard diet and water and administered 1 mL of vancomycin (VANC) via oral gavage. Vancomycin was administered at a dose of 2 mg/mouse every three days.MCD + VANC group (n = 7): Mice were fed an MCD diet and water and administered 1 mL of vancomycin via oral gavage. Vancomycin was administered at a dose of 2 mg/mouse every three days.

### 4.3. Diet

The MCD diet (obtained from Arden Research, Ankara, Türkiye) contains sucrose (40%) and fat (10%) and lacks methionine and choline. The MCD diet impairs hepatic mitochondrial β-oxidation and very-low-density lipoprotein synthesis, thereby causing steatosis and liver injury [9,13].

### 4.4. Vancomycin Dosing Rationale and Use

A murine oral dose of 50 mg/kg is commonly used for C. difficile treatment [28]. Based on the average weight (30–31 g), a bolus of 2 mg/mouse approximates this exposure. The q72h regimen was selected from prior studies showing microbiota modulation and reduced steatosis with intermittent dosing [29].

### 4.5. Sample Collection

At week 10, anesthesia was induced using intraperitoneal administration of ketamine (70 mg/kg) and xylazine (7 mg/kg). Blood was collected, and then animals were euthanized in accordance with ethical guidelines. Liver and cecal tissues were harvested; the liver was weighed and fixed in 10% neutral-buffered formalin, and cecal contents were snap-frozen at −80 °C for microbiota analysis.

### 4.6. Histochemical Evaluation

Fixed liver tissue samples (in 10% formaldehyde solution for 3 days) were embedded in paraffin blocks and divided into 5 µm sections. Slides were stained with hematoxylin–eosin, periodic acid–Schiff (PAS) and Masson’s trichrome. Semi-quantitative scoring followed Kleiner et al. [30]: steatosis (0–3), lobular inflammation (0–3), hepatocyte ballooning (0–2), and fibrosis (0–4). The NAFLD activity score (NAS) was calculated as the total of the steatosis, lobular inflammation, and hepatocyte ballooning scores (range: 0–8). A NAS ≥ 5 was considered “definite NASH,” NAS 3–4 as “possible NASH,” and NAS < 3 as “not compatible with NASH.” Fibrosis was scored separately according to the scale of Kleiner et al. (0 = no fibrosis; 4 = cirrhosis) [30].

Immunohistochemistry used the Avidin Biotin-Peroxidase (ABC) method on the 5 µm sections with anti-cytokeratin 18 (PA5-28279, Thermo, Waltham, MA, USA) and anti-ubiquitin (PA5-102555, Thermo) antibodies. After overnight incubation at +4 °C, biotinylated secondary antibody and enzyme-labeled (peroxidase) avidin-biotin complex were applied for 30 min each. Diaminobenzidine kit was used for visualization, and slides were counterstained with Mayer’s hematoxylin. Immunoreactivity was scored semi-quantitatively: 0 (–) = no staining, 1 (+) = slight staining, 2 (++) = moderate staining, and 3 (+++) = strong staining [17]. Microscopy has been performed with Olympus BX51.

### 4.7. Cytokine Analyses

Liver tissue IL-6, IL-8, and TGF-β1 levels were quantified using commercial ELISA kits (USCN (Dallas, TX, USA), for IL-6; Fine Test (Palm Coast, FL, USA), for IL-8 and TGF-β1), per manufacturers’ instructions; absorbance was read at 450 nm (BioTek ELx800 (Winooski, VT, USA)).

### 4.8. Microbiota Analysis

Stored cecal content was extracted with the High Pure PCR Template Preparation Kit (Roche, Basel, Switzerland) using bead-beating (0.1 mm glass or 1.0 mm zirconia beads) and proteinase K lysis, followed by column purification according to the manufacturer’s protocol. The 16S rRNA V3–V4 region was amplified with 314F-860R primers; libraries were prepared with Nextera XT and sequenced on an Illumina MiSeq (paired-end, 2 × 250 bp). Paired-end reads were processed in QIIME2. Denoising, quality trimming, and chimera removal were performed with DADA2 (q2-dada2) to infer amplicon sequence variants (ASVs). Taxonomy was assigned against the SILVA reference database (v138.1) and used for all primary analyses. QIIME2 artifacts were imported into R 4.1 and assembled into a phyloseq object. Alpha-diversity metrics (Shannon, Chao1, Simpson and observed ASVs) were computed with the microbiome/phyloseq packages. Between-group differences were assessed using Kruskal–Wallis tests with Benjamini–Hochberg FDR correction. Beta-diversity was calculated using Bray–Curtis, Jaccard, weighted and unweighted UniFrac analyses; ordination used PCoA. Group separation was tested by PERMANOVA. Intergroup *p* values were calculated using the Kruskal–Wallis test. Specific differences between groups were determined by differential abundance analysis using the Deseq2 R package. LEfSe identified discriminant taxa across groups (Kruskal–Wallis α = 0.05; LDA threshold specified in figure legends).

### 4.9. Statistical Analysis

Power analysis indicated that, with n = 7 per group, power is ~40–50% for a moderate effect (f = 0.4) but ≥80% for a large effect (f = 0.8), which aligns with prior NAFLD/NASH animal studies [18,19]. The data are expressed as mean ± standard deviation (SD) values. Normality (Shapiro–Wilk) and homoscedasticity (Levene) were assessed. Multiple comparisons were corrected using the Bonferroni/Benjamini–Hochberg/Tukey adjustment, and only FDR-adjusted *p*-values are reported. A *p*-value < 0.05 was considered statistically significant.

## 5. Conclusions

In conclusion, this study systematically compared the effects of oral vancomycin in the MCD model of NASH by integrating gut microbiota and liver histology outcomes using a full factorial design. We found that vancomycin treatment was associated with a decrease in markers of liver damage, but this improvement occurred alongside significant changes in the microbiota, characterized by a reduction in diversity and enrichment of pathogenic microorganisms. These findings highlight the dual effect of vancomycin, which alleviates liver damage on one hand, and on the other, triggers dysbiosis, underscoring the importance of combining antibiotic strategies with microbiota-supporting interventions (e.g., prebiotics, probiotics, or postbiotics). Due to the limited sample size and power, our results should be interpreted with caution, but they highlight potential trade-offs in gut–liver axis modulation and provide a foundation for more comprehensive studies, including bile acid profiles, LPS, and permeability tests.

## Figures and Tables

**Figure 1 ijms-26-08616-f001:**
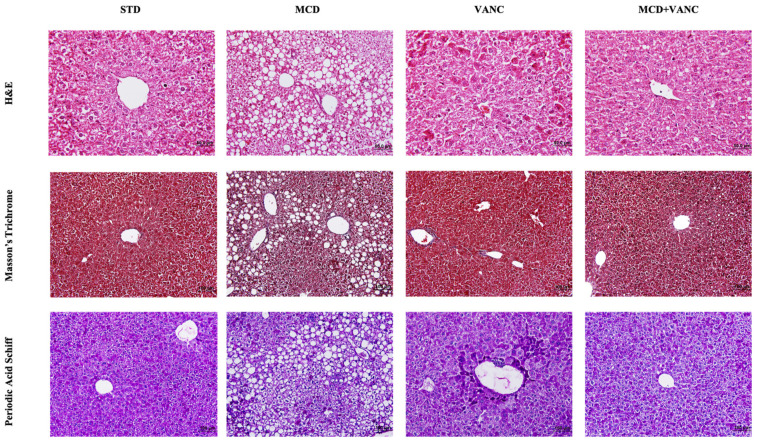
In the MCD group, extensive macrovesicular steatosis was observed. In contrast, the MCD + VANC group demonstrated microvesicular steatosis, suggesting a reduction in steatosis severity compared to the MCD group. Both the MCD and MCD + VANC groups exhibited dilated central veins and inflammatory infiltrates. The images were captured at ×40 magnification and stained with hematoxylin and eosin (H&E), Masson’s trichrome, and periodic acid–Schiff (PAS).

**Figure 2 ijms-26-08616-f002:**
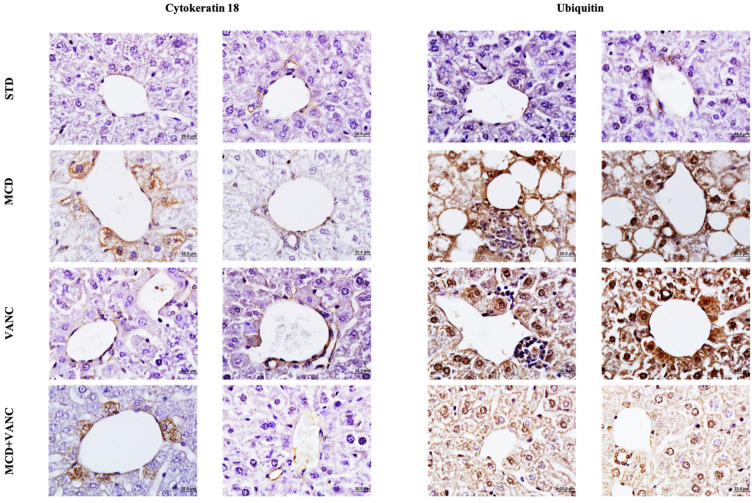
Histomorphological and immunohistochemical scoring of liver tissue. Both ubiquitin and cytokeratin immunoreactivity were elevated in the liver tissue of the MCD group. Immunoreactivity was significantly reduced in the MCD + VANC group, as shown in the graphs (non-significant and *p* < 0.001).

**Figure 3 ijms-26-08616-f003:**
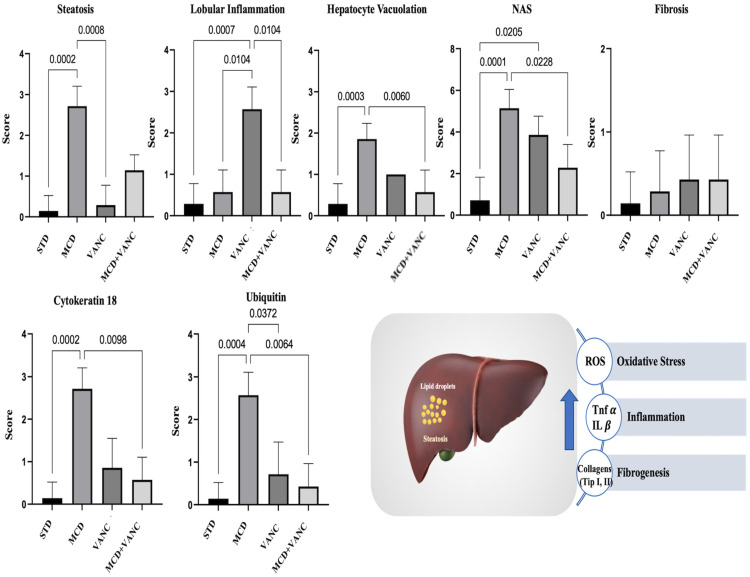
Cytokeratin 18 immunoreactivity was observed in hepatocytes adjacent to the central vein, with low reactivity intensity in the portal area in all the groups. Ubiquitin immunoreactivity was notably severe in the central vein and portal areas of the MCD and VANC groups. The MCD + VANC group showed decreased reactivity compared to the MCD group.

**Figure 4 ijms-26-08616-f004:**
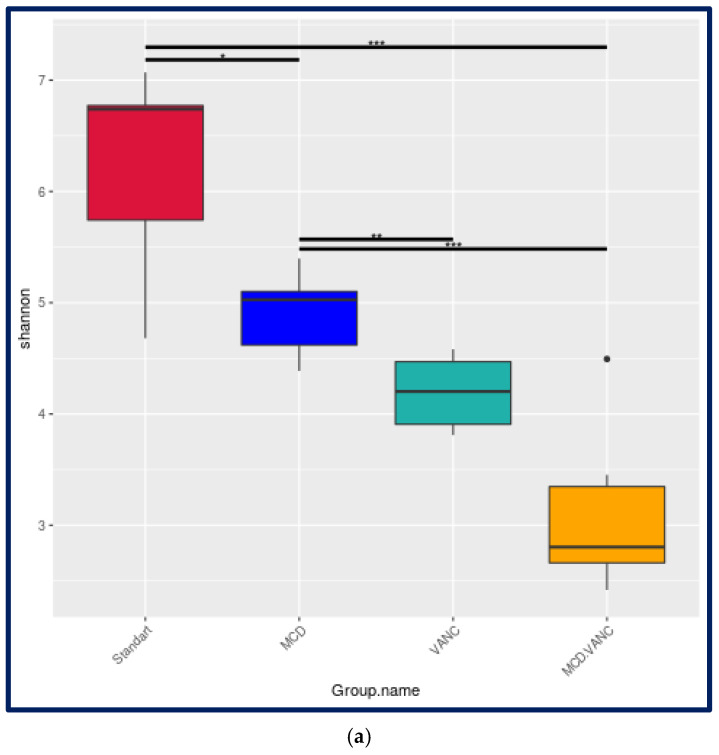
(**a**) Cecal microbiota α-diversity (Shannon index) across experimental groups. Standard diet (STD), methionine–choline-deficient diet (MCD), STD + vancomycin (VANC; 2 mg/mouse by oral gavage every 3 days), and MCD + vancomycin (MCD + VANC). Boxplots show median (center line), interquartile range (box), and 1.5 × IQR whiskers; dots denote outliers. Statistics: Kruskal–Wallis followed by Dunn’s post hoc test with Benjamini–Hochberg correction. Asterisks above brackets indicate significance (* <0.05, ** <0.01, *** <0.001). Abbreviations: STD, standard diet; MCD, methionine–choline-deficient diet; VANC, vancomycin. (**b**) Cecal microbiota richness (Chao1 index) across experimental groups. Standard diet (STD), methionine–choline-deficient diet (MCD), STD + vancomycin (VANC; 2 mg/mouse by oral gavage every 3 days), or MCD + vancomycin (MCD + VANC). Boxplots display median, interquartile range (IQR), and 1.5 × IQR whiskers; dots denote outliers. Statistics: Kruskal–Wallis with Dunn’s post hoc test (Benjamini–Hochberg correction). Pairwise comparisons (adjusted P). Abbreviations: STD, standard diet; MCD, methionine–choline-deficient diet; VANC, vancomycin.

**Figure 5 ijms-26-08616-f005:**
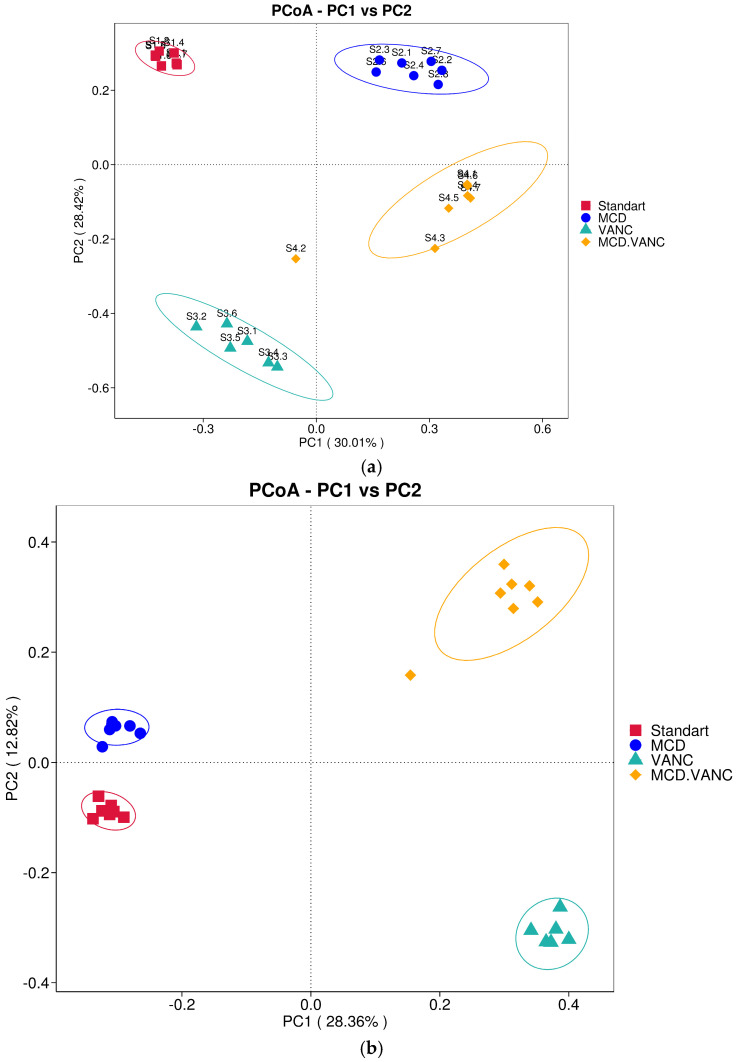
(**a**) Principal coordinates analysis (PCoA) of Bray–Curtis dissimilarities for cecal microbiota. Each point is one mouse (n = 7/group): standard diet (STD, red squares); methionine–choline-deficient diet (MCD, blue circles); STD + vancomycin (VANC, teal triangles); and MCD + vancomycin (MCD + VANC, orange diamonds). Ellipses denote 95% confidence intervals around group centroids. PC1 and PC2 explain 30.01% and 28.42% of the variance, respectively. Distances were computed from relative-abundance ASV tables. Group separation was tested by pairwise PERMANOVA; *p* values: STD vs. VANC, 0.001; MCD vs. STD, 0.002; MCD vs. VANC, 0.003; MCD + VANC vs. STD, 0.001; MCD + VANC vs. VANC, 0.003; MCD + VANC vs. MCD, 0.003. (**b**) Principal coordinates analysis (PCoA) of Jaccard dissimilarities for cecal microbiota. Each point is one mouse (n = 7/group): standard diet (STD, red squares), methionine–choline-deficient diet (MCD, blue circles), STD + vancomycin (VANC, teal triangles), and MCD + vancomycin (MCD + VANC, orange diamonds). Ellipses denote 95% confidence intervals around group centroids. PC1 and PC2 explain 28.3% and 12.8% of the variance, respectively. Distances were computed from relative-abundance ASV tables.

**Figure 6 ijms-26-08616-f006:**
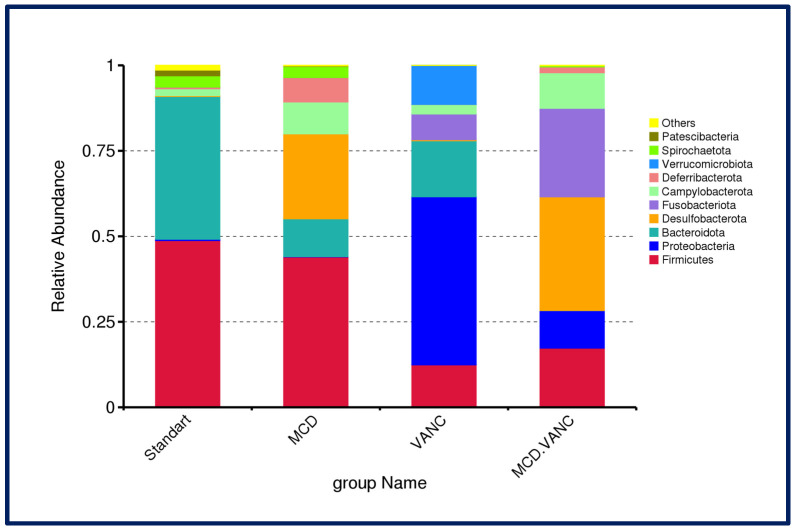
Cecal microbiota composition at the phylum level across experimental groups. Stacked bars depict the relative abundance of bacterial phyla in male C57BL/6J mice (n = 7/group) after 10 weeks on standard diet (STD), methionine–choline-deficient diet (MCD), STD + vancomycin (VANC; 2 mg/mouse by oral gavage every 3 days), and MCD + vancomycin (MCD + VANC). Phyla with low mean abundance were pooled as “Others”.

**Figure 7 ijms-26-08616-f007:**
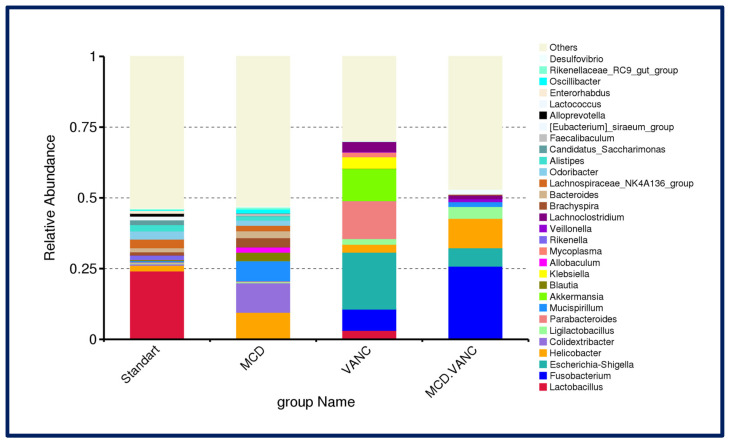
Genus-level cecal microbiota composition across experimental groups. Stacked bars show the mean relative abundance of bacterial genera in male C57BL/6J mice (n = 7/group) after 10 weeks on standard diet (STD), methionine–choline-deficient diet (MCD), STD + vancomycin (VANC; 2 mg/mouse by oral gavage every 3 days), and MCD + vancomycin (MCD + VANC). Genera with low mean abundance were pooled as “Others”.

**Figure 8 ijms-26-08616-f008:**
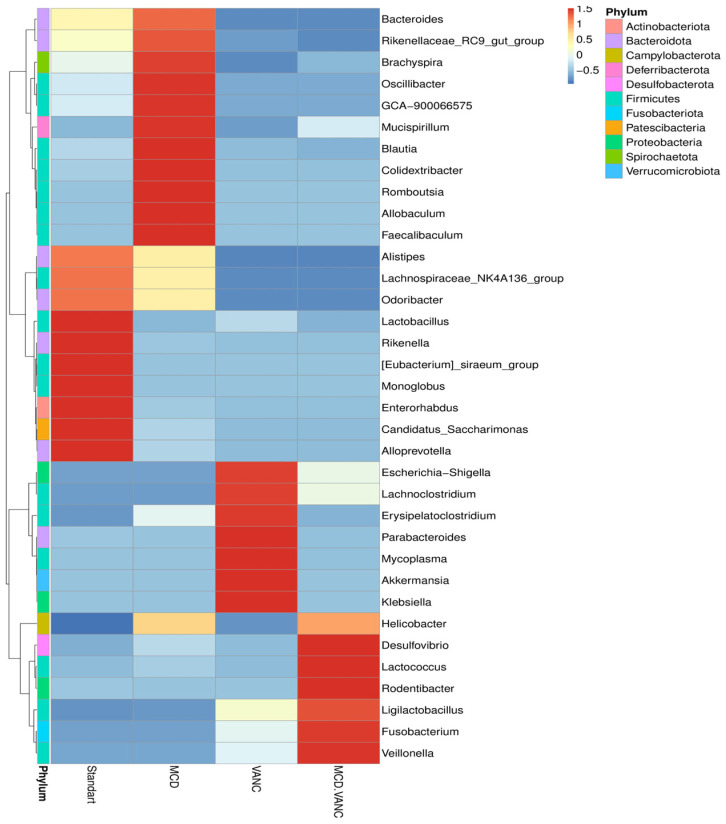
Genus-level heatmap of the cecal microbiota across diet/antibiotic groups. Columns show group means for standard chow (STD), methionine–choline-deficient diet (MCD), STD + oral vancomycin (VANC), and MCD + vancomycin (MCD + VANC; labeled “MCD.VANC” in the plot). Rows are genera (right), limited to the most abundant taxa across groups. Cell colors denote centered-and-scaled abundance for each genus (higher than that genus’ mean = red; lower = blue). Rows are clustered by similarity (dendrogram, (left)). The color strip beside the rows annotates phylum identity (legend, (right)).

**Figure 9 ijms-26-08616-f009:**
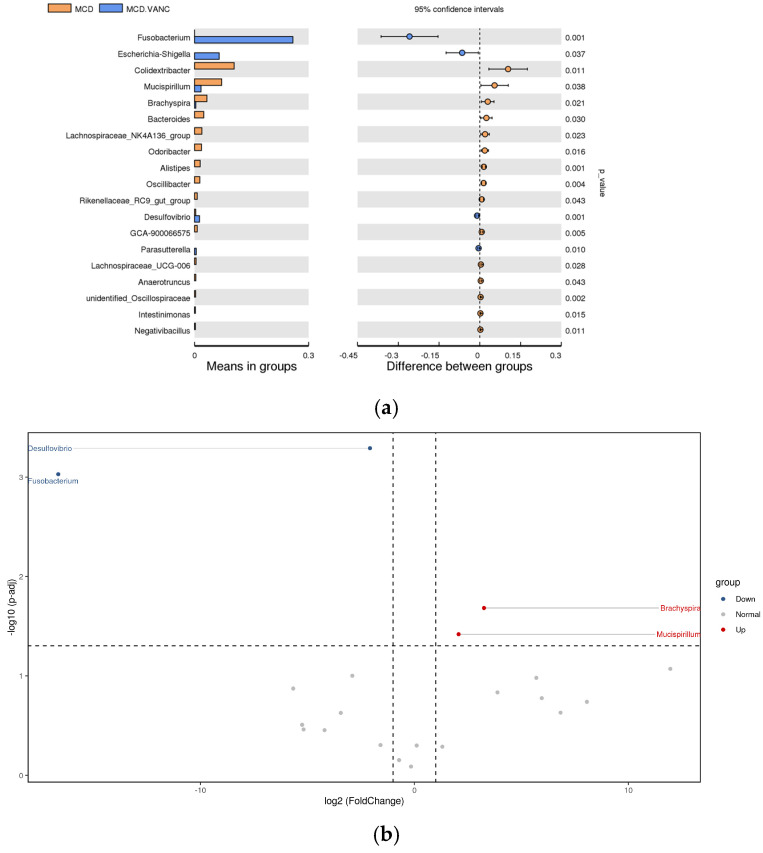
(**a**) Differential abundance analysis of the genera between MCD and MCD + VANC groups. Orange: MCD (methionine–choline-deficient) diet; blue: methionine–choline-deficient (MCD) diet with vancomycin treatment. Left panel—Means in groups: Horizontal bars represent the mean relative abundance of each genus in the two groups. Each taxon is listed on the *Y*-axis. The length of the bar reflects the average abundance level in each group. Right panel—Difference between groups: Circles represent the mean difference in relative abundance between groups (MCD.VANC-MCD). Horizontal error bars show the 95% confidence intervals. If the confidence interval does not cross zero, the difference is considered statistically significant. The dotted vertical line at 0 indicates no difference between groups. *p*-values are displayed to the right of each row. They indicate the statistical significance of the difference between groups for each genus. Genera with *p* < 0.05 are considered significantly different. (**b**) Differential abundance analysis of the genera was carried out using ANCOM-BC for unpaired comparisons between MCD vs. MCD + VANC groups. Blue dots: most significantly overrepresented genera in the MCD + VANC group. Red dots: most significantly overrepresented genera in the MCD group.

**Figure 10 ijms-26-08616-f010:**
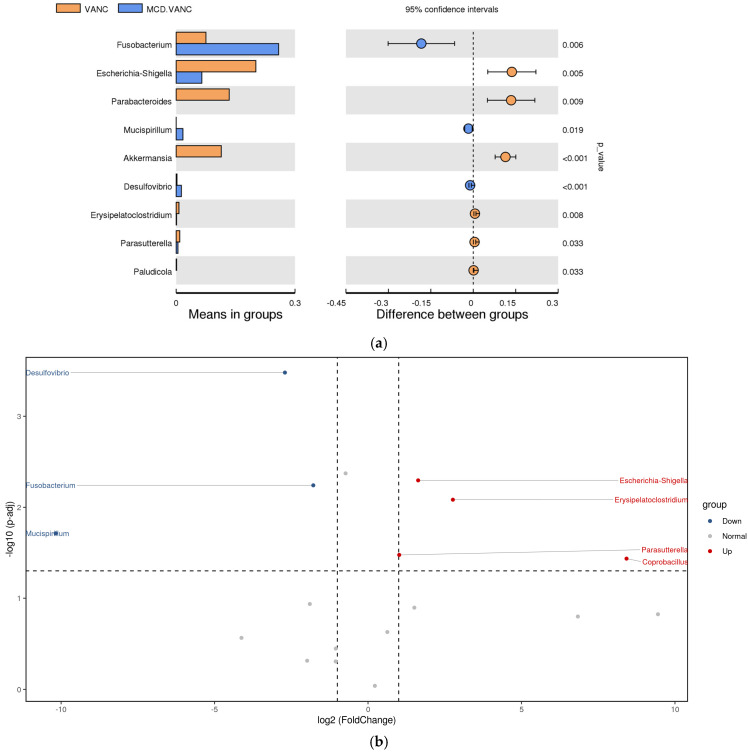
(**a**) Differential abundance analysis of the genera between VANC and MCD + VANC groups. Orange: vancomycin; blue: methionine–choline-deficient (MCD) diet with vancomycin treatment. Left panel—Means in groups: Horizontal bars represent the mean relative abundance of each genus in the two groups. Each taxon is listed on the *Y*-axis. The length of the bar reflects the average abundance level in each group. Right panel—Difference between groups: Circles represent the mean difference in relative abundance between groups (MCD.VANC-VANC). Horizontal error bars show the 95% confidence intervals. If the confidence interval does not cross zero, the difference is considered statistically significant. The dotted vertical line at 0 indicates no difference between groups. *p*-values are displayed to the right of each row. They indicate the statistical significance of the difference between groups for each genus. Genera with *p* < 0.05 are considered significantly different. (**b**) Differential abundance analysis of the genera was carried out using ANCOM-BC for unpaired comparisons between VANC vs. MCD + VANC groups. Blue dots: most significantly overrepresented genera in the MCD + VANC group. Red dots: most significantly overrepresented genera in VANC group.

**Figure 11 ijms-26-08616-f011:**
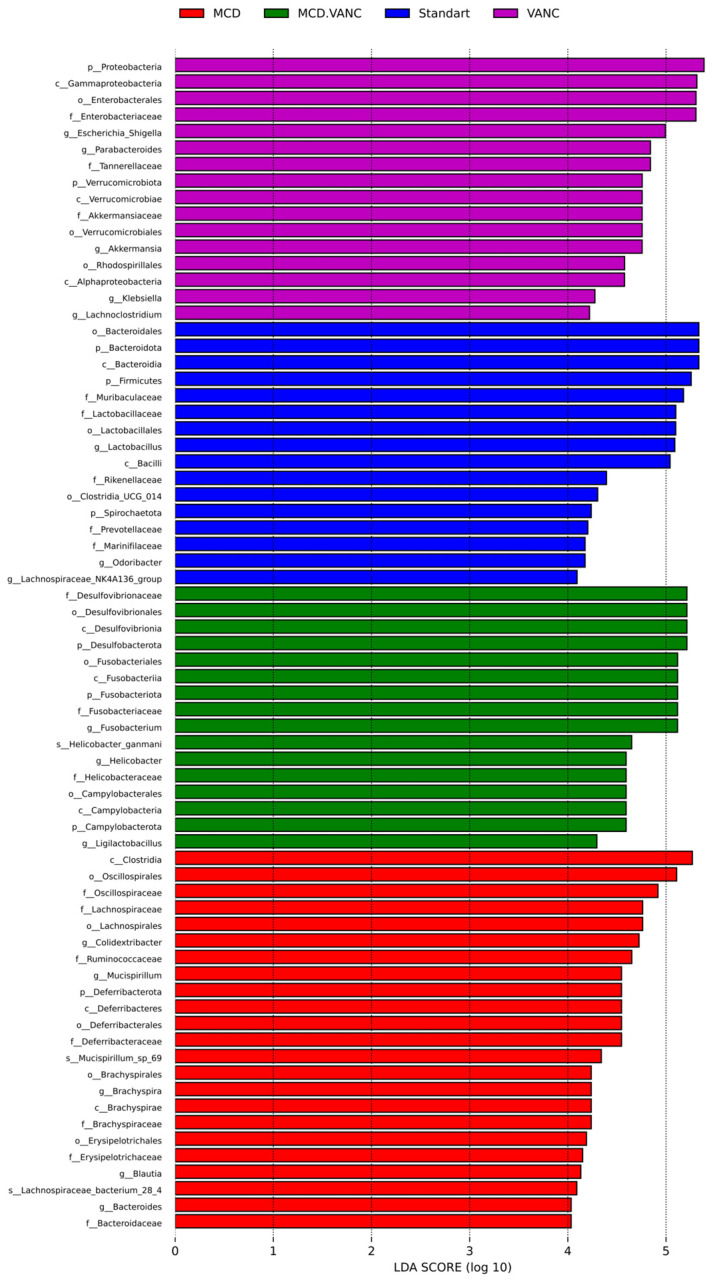
LEfSe multiclass analysis identifies group-specific bacterial biomarkers across the four study arms. Cecal 16S profiles from standard diet (blue), MCD (red), VANC (purple), and MCD + VANC (green) mice (n = 7/group) were compared by LEfSe (Kruskal–Wallis α = 0.05; pairwise Wilcoxon α = 0.05; LDA score threshold = 4.0). Bars show taxa that discriminate each group, ordered by linear discriminant analysis (LDA) score (log10).

**Table 1 ijms-26-08616-t001:** Hepatic cytokine levels of the study groups.

Group	IL-6 (pg/mg)	IL-8 (pg/mg)	TGF-β1 (ng/mg)
STD	20.6 ± 1.3	2833 ± 544	2973 ± 632
MCD	15.8 ± 0.9	1660 ± 404	2089 ± 523
VANC	17.8 ± 0.76	1953 ± 384	2114 ± 334
MCD + VANC	17.9 ± 2.6	999 ± 124 *	1045 ± 95.8 *

Values of the following are expressed as mean ± SD: IL-6 and IL-8 (pg/mg), and TGF-β1 (ng/mg). Asterisks indicate significant differences versus the STD group after multiple comparison correction. Exact *p* for MCD + VANC vs. STD: IL-8 *p* = 0.003; TGF-β1 *p* = 0.006. All other pairwise comparisons (including MCD vs. STD and MCD vs. MCD + VANC) were not significant. Abbreviations: STD, standard diet; MCD, methionine–choline-deficient diet; VANC, vancomycin.

## Data Availability

The data collected during this study are available from the corresponding author upon reasonable request (after publication).

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
