# Peer review of "Hepatic Histopathological Benefit, Microbial Cost: Oral Vancomycin Mitigates Non-Alcoholic Fatty Liver Disease While Disrupting the Cecal Microbiota"

_ijms, 2025, doi:10.3390/ijms26178616_

Round 1
Reviewer 1 Report
Comments and Suggestions for Authors
The paper examines if oral vancomycin alters the severity of NASH and cecal microbiota composition in male C57BL/6J mice receiving a methionine–choline-deficient (MCD) diet. Animals were randomized into four groups (STD, MCD, VANC, MCD+VANC; n=7/group) for 10 weeks (2 mg/mouse every 72 hours by mouth) and histopathological changes, CK18/ubiquitin, cytokines (IL-6, IL-8, TGF-β1), and 16S rRNA microbiota (QIIME2) were determined. MCD caused characteristic NASH features; in addition to vancomycin, there was a trend towards microvesicular steatosis and reduction in CK18/ubiquitin.
The paper in my view adds a distinct, factorial representation of the "diet × antibiotic" interaction within the gut–liver axis: vancomycin within the setting of an MCD lowers markers of hepatocellular damage and pro-fibrotic signals (CK18, ubiquitin, IL-8, TGF-β1) but causes a low-rich/low-uniformity state with pathogen amplification—a clear therapeutic "trade-off" that must be weighed in targeting the microbiome for NAFLD/NASH.
To my view, the article needs some major changes.
1. The Results section reports a difference in "IL-14" between groups, while Table 1 and the Methods section report measurement of IL-6, IL-8, and TGF-β1. Given the specific p-values given for IL-8 and TGF-β1, it would be clearer to replace "IL-14" with the correct cytokine, presumably IL-8, to be consistent between text and table.
2. Certain sentences mix "not significant" and "p<0.001" in the same sentence for the same result in the figures, which is perplexing. It may be indicated which pair of comparison is significant and a consistent formatting may be used (e.g., "p = 0.003", "p" in lowercase throughout, with reporting corrections for multiple testing where relevant).
3. "Chao 1" and "Chao1" are used alternately. Consistent usage (Chao1) and parallelism of units/statistics (mean ± SD, corrected p-values) would enhance the readability of Figures 4a–4b and the paragraph.
4. The Methods state primary assignment to Greengenes and "confirmation" with SILVA. With SILVA being updated, it would be better to utilize it as the primary source and move Greengenes to a sensitivity analysis, with clear reporting of the discrepancies in the results if any.
5. Discussion mentions that "body weight did not change", but no curves/tests are shown. It would be helpful to include weight course graphs by animal/group (with statistical control stated in the protocol), at least as Supplementary Material, to back up the conclusion.
6. Dose justification (2 mg/mouse ≈ 50 mg/kg, q72h) is made in Methods; a brief mention of the mg/kg equivalent in the Abstract would allow for ready comparison with other literature and protocols.
7. The text contains the terms "MCD+VANC" and "MCD-VANC", as well as citations such as "Figure 2-3". Following a consistent convention, for example using "MCD+VANC" and "Figures 2–3" throughout, would greatly enhance consistency overall.
8. Units (IL-6/IL-8 in pg/mg and TGF-β1 in ng/mg) need to be reaffirmed and it needs to be clearly indicated in the caption that the asterisks are for comparisons with STD, along with whether multiple comparison correction was done on the p-values in the table.
9. The paper itself mentions that n=7/group provides ~40–50% power for moderate effects. It would be helpful to expressly "lock in" the conclusions under this limitation, with cautious wording (e.g. "compatible with reduction"), and to indicate for which analyses FDR was used.
10. Since it is given "available upon request", it would be strongly assisting to submit the raw 16S fastq and metadata in the SRA (with accession), as well as scripts/QIIME2 artifacts, to increase transparency and reproducibility.
11. Some errors are noted ("Stoted", "nd slides", "have been performed"). A rigorous linguistic editing (grammar/punctuation/syntax) would enhance readability, particularly in the Methods and captions.
12. The use of the word "driver" to describe IL-8 as a "driver" of fibrosis perhaps sounds a bit prescriptive. It would be better to state it as "associated with/correlated with" inflammation/fibrosis, which correlates the language to the type of data in the current study.
13. Given that the authors have identified the absence of bile acids, lipopolysaccharides (LPS), and permeability markers as a limitation, a brief suggestion for the inclusion of these components (e.g., bile acid profiles and FITC-dextran) in the future would provide more specific direction.
14. Escherichia–Shigella, Klebsiella, Fusobacterium, etc. increases are reported. The inclusion of a heatmap with log2-fold changes (Deseq2) and FDR-corrected p-values for pairwise comparisons would help the reader connect the results to the hepatic endpoints.
Author Response
Dear Editor and Reviewer’s;
We would like to thank the referees and editors for their comments on our article.
You will find our point-by-point responses to all suggestions and proposals (yellow-colored).
We also added a new Figure for heatmap of genus and also one Supplementary file for body weight.
Our manuscript latest version have been edited for language by MDPI language editing.
Reviewer-1
The paper examines if oral vancomycin alters the severity of NASH and cecal microbiota composition in male C57BL/6J mice receiving a methionine–choline-deficient (MCD) diet. Animals were randomized into four groups (STD, MCD, VANC, MCD+VANC; n=7/group) for 10 weeks (2 mg/mouse every 72 hours by mouth) and histopathological changes, CK18/ubiquitin, cytokines (IL-6, IL-8, TGF-β1), and 16S rRNA microbiota (QIIME2) were determined. MCD caused characteristic NASH features; in addition to vancomycin, there was a trend towards microvesicular steatosis and reduction in CK18/ubiquitin.
The paper in my view adds a distinct, factorial representation of the "diet × antibiotic" interaction within the gut–liver axis: vancomycin within the setting of an MCD lowers markers of hepatocellular damage and pro-fibrotic signals (CK18, ubiquitin, IL-8, TGF-β1) but causes a low-rich/low-uniformity state with pathogen amplification—a clear therapeutic "trade-off" that must be weighed in targeting the microbiome for NAFLD/NASH.
Thank you for positive comments about our article.
To my view, the article needs some major changes.
- The Results section reports a difference in "IL-14" between groups, while Table 1 and the Methods section report measurement of IL-6, IL-8, and TGF-β1. Given the specific p-values given for IL-8 and TGF-β1, it would be clearer to replace "IL-14" with the correct cytokine, presumably IL-8, to be consistent between text and table.
We excluded the data about the serum IL-14 levels from the Results section, this is a typographical error, and we don’t have mention serum IL-14 in the Methods section too.
Certain sentences mix "not significant" and "p<0.001" in the same sentence for the same result in the figures, which is perplexing. It may be indicated which pair of comparison is significant and a consistent formatting may be used (e.g., "p = 0.003", "p" in lowercase throughout, with reporting corrections for multiple testing where relevant).
According to your comments, we changed as: “Asterisks indicate significant differences versus the STD group after multiple comparison correction. Exact p for MCD+VANC vs. STD: IL-8 p = 0.003; TGF-β1 p = 0.006. All other pairwise comparisons (including MCD vs. STD and MCD vs. MCD+VANC) were not significant. Abbreviations: STD, standard diet; MCD, methionine–choline-deficient diet; VANC, vancomycin.”
"Chao 1" and "Chao1" are used alternately. Consistent usage (Chao1) and parallelism of units/statistics (mean ± SD, corrected p-values) would enhance the readability of Figures 4a–4b and the paragraph.
We changed as Chao1, as you recommended.
The Methods state primary assignment to Greengenes and "confirmation" with SILVA. With SILVA being updated, it would be better to utilize it as the primary source and move Greengenes to a sensitivity analysis, with clear reporting of the discrepancies in the results if any.
We checked our analysis and we changed as: “Taxonomy was assigned against the SILVA reference database (v138.1) and used for all primary analyses.”
Discussion mentions that "body weight did not change", but no curves/tests are shown. It would be helpful to include weight course graphs by animal/group (with statistical control stated in the protocol), at least as Supplementary Material, to back up the conclusion.
We changed as “Across the 10-week study, no difference in body weight was observed among groups (p > 0.05) (Supplementary Figure 5)” and add new Supplementary Figure 5.
Dose justification (2 mg/mouse ≈ 50 mg/kg, q72h) is made in Methods; a brief mention of the mg/kg equivalent in the Abstract would allow for ready comparison with other literature and protocols.
Corrected according to your comments.
The text contains the terms "MCD+VANC" and "MCD-VANC", as well as citations such as "Figure 2-3". Following a consistent convention, for example using "MCD+VANC" and "Figures 2–3" throughout, would greatly enhance consistency overall.
We thank the reviewer for this observation. We have carefully revised the manuscript to ensure consistent terminology and citation formatting throughout. Specifically, we now use “MCD+VANC” uniformly (instead of alternating with “MCD-VANC”), and all figure citations follow the convention “Figures” if needed.
- Units (IL-6/IL-8 in pg/mg and TGF-β1 in ng/mg) need to be reaffirmed and it needs to be clearly indicated in the caption that the asterisks are for comparisons with STD, along with whether multiple comparison correction was done on the p-values in the table.
We added our statistical comparison and added information at the end of the table.
9. The paper itself mentions that n=7/group provides ~40–50% power for moderate effects. It would be helpful to expressly "lock in" the conclusions under this limitation, with cautious wording (e.g. "compatible with reduction"), and to indicate for which analyses FDR was used.
We thank the reviewer for highlighting this important point. We acknowledge that with n = 7 per group, the statistical power for detecting moderate effects is limited (~40–50%). To address this, we have revised the Discussion and Conclusions to explicitly “lock in” our interpretations under this limitation, using more cautious wording (e.g., “compatible with a reduction” rather than “significantly reduced”).
Furthermore, we have clarified in the Methods and figure/table legends which analyses were subjected to false discovery rate (FDR) correction, and only FDR-adjusted p-values are interpreted for multiple comparisons.
Since it is given "available upon request", it would be strongly assisting to submit the raw 16S fastq and metadata in the SRA (with accession), as well as scripts/QIIME2 artifacts, to increase transparency and reproducibility.
We still collect the data for deposition and will plan to share with open available system, however regarding the regulation is not possible now.
Some errors are noted ("Stoted", "nd slides", "have been performed"). A rigorous linguistic editing (grammar/punctuation/syntax) would enhance readability, particularly in the Methods and captions.
Corrected and all manuscript revised by professional language editing.
- The use of the word "driver" to describe IL-8 as a "driver" of fibrosis perhaps sounds a bit prescriptive. It would be better to state it as "associated with/correlated with" inflammation/fibrosis, which correlates the language to the type of data in the current study.
We thank the reviewer for this valuable suggestion. We agree that describing IL-8 as a “driver” of fibrosis may overstate the causal inference from our data. We have revised the text to use more precise and appropriate language, replacing “driver” with “associated with” or “correlated with,” to better reflect the observational nature of our findings and align with the type of data presented.
- Given that the authors have identified the absence of bile acids, lipopolysaccharides (LPS), and permeability markers as a limitation, a brief suggestion for the inclusion of these components (e.g., bile acid profiles and FITC-dextran) in the future would provide more specific direction.
We appreciate the reviewer’s insightful comment. We agree that the absence of bile acids, lipopolysaccharides (LPS), and permeability markers is a limitation of the present study. We have now revised the Discussion to include a brief statement highlighting the importance of these components and suggesting their assessment in future work, for example through bile acid profiling and FITC-dextran assays for intestinal permeability.
Escherichia–Shigella, Klebsiella, Fusobacterium, etc. increases are reported. The inclusion of a heatmap with log2-fold changes (Deseq2) and FDR-corrected p-values for pairwise comparisons would help the reader connect the results to the hepatic endpoints.
According to your comments we added new heat map for genus levels as Figure 8

Reviewer 2 Report
Comments and Suggestions for Authors
The paper „Hepatic Histopathological Benefit, Microbial Cost: Oral Vancomycin Mitigates Non-Alcoholic Fatty Liver Disease While Disrupting the Cecal Microbiota” presents the original results of an experimental research conducted on animal model. As such it is worthy of attention in the area of hepatic health and both the mechanisms behind the non-alcoholic fatty liver disease and the potential therapeutic approach given in this paper. Some changes should be taken under consideration in order to improve the paper. I would particularly like to compliment the design of the study that included a group of animals treated with vancomycin only in order to have a control group of the possible hepatotoxic effects of the drug per se.
- The abstract is too long. It should have 250 words and shorter conclusion with more highlighted results.
- There is no clearly defined objectives in this paper. So at the end of the introduction part there is a clear need for an objectives.
- The discussion requires explanation of the obtained results besides the comparison and comments about other researches.
- How do you explain the changes of interleukin levels between the groups? Explain the results of Table 1. Why interleukin levels are affected by vancomycin, how it affects the whole liver health?
- Please comment this results together with the results given in Figure 3 in which the highest level of inflammation was registered in VAC group.
- Which bacterial strains were most affected in VAC group? Why? Is this beneficial rather than harmful for the liver health. Connect it with the results obtained in Figure 3 and may be with the interplay of dysbiosis the leaky gut effect and the endotoxins influence of liver health.
- The Conclusion part needs to be more specific. It is too general.
- Figure 3 should be larger since it gives a lot of results which significance seem to be minimized.
Author Response
Reviewer-2
The paper „Hepatic Histopathological Benefit, Microbial Cost: Oral Vancomycin Mitigates Non-Alcoholic Fatty Liver Disease While Disrupting the Cecal Microbiota” presents the original results of an experimental research conducted on animal model. As such it is worthy of attention in the area of hepatic health and both the mechanisms behind the non-alcoholic fatty liver disease and the potential therapeutic approach given in this paper. Some changes should be taken under consideration in order to improve the paper. I would particularly like to compliment the design of the study that included a group of animals treated with vancomycin only in order to have a control group of the possible hepatotoxic effects of the drug per se.
Thank you for your positive opinion about our manuscript.
- The abstract is too long. It should have 250 words and shorter conclusion with more highlighted results.
We revised the Abstract according to your comments and now include 261 words
- There is no clearly defined objectives in this paper. So at the end of the introduction part there is a clear need for an objectives.
Thank you for your comments. We added our Objectives as: “Therefore, the objective of this study was to investigate the effects of superimposed oral vancomycin on histopathological changes, including liver fibrosis, inflammatory markers, and gut microbiota composition in an experimental NAFLD model.”
- The discussion requires explanation of the obtained results besides the comparison and comments about other researches.
We thank the reviewer for this helpful suggestion. In response, we have revised the Discussion to expand on the explanation of our own results, in addition to comparing them with prior work. Specifically, we now more clearly interpret the shifts in steatosis pattern, cytokine changes, and microbiota alterations within the context of our experimental design. We also added explanatory links between Figures 3, Table 1, and the observed dysbiosis, highlighting the trade-off between reduced hepatic injury markers and increased inflammatory pressure from the gut.
- How do you explain the changes of interleukin levels between the groups? Explain the results of Table 1. Why interleukin levels are affected by vancomycin, how it affects the whole liver health?
- Please comment this results together with the results given in Figure 3 in which the highest level of inflammation was registered in VAC group.
- Which bacterial strains were most affected in VAC group? Why? Is this beneficial rather than harmful for the liver health. Connect it with the results obtained in Figure 3 and may be with the interplay of dysbiosis the leaky gut effect and the endotoxins influence of liver health.
Thank you for your important comments. Regarding your comments (3-6), we added a paragraph into the Discussion as “The reduction in IL-6 and IL-8 in the MCD-VANC groups suggests a dampening of hepatic inflammation, while decreased TGF-β1 levels are compatible with reduced fi-brogenesis. This cytokine profile parallels the histological improvements observed. However, vancomycin induces profound dysbiosis, including the enrichment of patho-bionts, which means that the protective cytokine changes may coexist with an increased risk of microbial translocation and endotoxin-driven injury. This duality underscores that while vancomycin can reduce inflammatory mediators, its long-term benefit for liver health is uncertain unless paired with microbiota-supportive strategies. The apparent paradox of lower IL-6/IL-8/TGF-β1 levels alongside the highest inflammatory profile (lobular inflammation) in the VANC alone group underscores the complex interplay between the microbiota and host responses. Vancomycin depleted key Gram-positive commensals, leading to a significant decrease in Lactobacillus and a reduction in Bac-teroides. Vancomycin promotes blooms of Escherichia-Shigella and Parabacteroides. These shifts align with the Proteobacteria-dominated phylum profile and the marked loss of taxonomic richness observed after vancomycin treatment. One possible explanation is that while vancomycin reduces direct hepatic inflammatory signaling, the resulting dysbiosis promotes leaky gut and endotoxin exposure, sustaining systemic and mucosal inflammation. Thus, vancomycin’s net effect reflects both transient liver improvement and heightened inflammatory pressure from the gut environment.”
- The Conclusion part needs to be more specific. It is too general.
Thank you for your comments. We changed the Conclusion according to your comments as: In conclusion, this study systematically compared the effects of oral vancomycin in the MCD model of NASH by integrating gut microbiota and liver histology outcomes using a full factorial design. We found that vancomycin treatment was associated with a decrease in markers of liver damage, but this improvement occurred alongside significant changes in the microbiota, characterized by a reduction in diversity and enrichment of pathogenic microorganisms. These findings highlight the dual effect of vancomycin, which alleviates liver damage on one hand, and on the other, triggers dysbiosis, underscoring the im-portance of combining antibiotic strategies with microbiota-supporting interventions (e.g., prebiotics, probiotics, or postbiotics). Due to the limited sample size and power, our results should be interpreted with caution, but they highlight potential trade-offs in gut–liver axis modulation and provide a foundation for more comprehensive studies, in-cluding bile acid profiles, LPS, and permeability tests.
- Figure 3 should be larger since it gives a lot of results which significance seem to be minimized.
We enlarged the Figure 3 as possible.

Round 2
Reviewer 1 Report
Comments and Suggestions for Authors
The revised version of the manuscript finds me in complete agreement. The authors made all corrections and answered all issues.
Reviewer 2 Report
Comments and Suggestions for Authors
The authors have included all the required changes from the Review Report in the reviewed version of the manuscript. I have no further comments.